# SA-Learner: Surrogate-Assisted Meta-Learner with Missing Outcomes

## Abstract

Estimating heterogeneous treatment effects is essential for personalized decision-making across various applications. While existing methods primarily focus on the conditional average treatment effect (CATE) for fully observed outcomes, real-world data often suffer from missingness. Direct CATE estimation using only complete cases can introduce bias and reduce efficiency. To address these challenges, we propose the Surrogate-Assisted Learner (SA-learner), which leverages surrogate outcomes—auxiliary variables expected to predict the effect of a treatment on the primary outcome and is more readily observed—to improve CATE estimation. The SA-learner enjoys double robustness, ensuring consistent CATE estimates even under misspecification of certain nuisance functions. We also establish its convergence rate, requiring only slower-rate convergence of nuisance function estimators without restrictive model assumptions. This property enables flexible implementation using off-the-shelf machine learning algorithms. Extensive experiments on synthetic data further demonstrate effectiveness of the proposed method.

## 1 Introduction

Heterogeneous treatment effect (HTE), studying the effect of a treatment or intervention on an outcome of interest across different subgroups or individuals within a population, is crucial for personalized decision-making in fields such as medicine (Collins & Varmus, 2015; Kent et al., 2020), economics (Heckman & Vytlacil, 2005; Bitler et al., 2006), and policy design (Ludwig et al., 2011; GREEN & KERN, 2012). A key focus in HTE analysis is the Conditional Average Treatment Effect (CATE), which measures the expected treatment effect given a set of covariates. Most existing work, including Bayesian methods (Hill, 2011; Alaa & van der Schaar, 2017; Hahn et al., 2020), tree-based approaches (Athey & Imbens, 2016; Wager & Athey, 2018), neural networks (Johansson et al., 2016; Shalit et al., 2017; Yoon et al., 2018; Shi et al., 2019; Hassanpour & Greiner, 2020), and meta-learners (Künzel et al., 2019; Nie & Wager, 2020; Kennedy, 2023), assumes complete response data for CATE estimation. However real-world scenarios often involve missing responses due to factors such as nonresponse to survey questions, recording errors, and loss to follow up (Little & Rubin, 2019). To address this challenge, we propose a novel method that introduces surrogate outcomes for missing response settings for the estimation of CATE. Our approach advances HTE analysis by providing a surrogate-assisted framework that improves efficiency and reduces estimation bias in the presence of missing data.

A central challenge in causal inference is the fundamental problem that, at the individual level, one cannot observe the outcomes of both treatment and control arms simultaneously (Holland, 1986). In this paper, we consider a even more challenging scenario: the outcomes of some individuals are possibly missing. A naive approach is to delete the individuals with missing outcome, but this will lead to efficiency loss and may trigger estimation bias (Hogan et al., 2004), especially when the missing is informative. Recently, Chakrabortty & Dai (2024) considered the settings of missing completely at random (MCAR) and Zhang et al. (2023) studied the settings of the missing at random (MAR), both show that incorporating unlabeled data could improve efficiency. We show in this paper that it is possible to provide further improvements Under the MAR settings, if we incorporate some auxiliary variables.

In practice, the primary outcome of interest may be missing as its collection can often be costly, impractical, or infeasible. However, some auxiliary variables, that may be highly related to the outcome, are easier to access. For instance, blood pressure or body weight are strongly related to cardiovascular disease and are easy and less expensive to collect. These variables are therefore frequently used in evaluating the effectiveness of new drug treatments targeting cardiovascular risk factors (Prentice, 1989; Fleming & DeMets, 1996; Psaty et al., 1999). These auxiliary variables or intermediate outcomes are known as surrogates outcomes and have been used to replace the missing primary outcome in recent causal inference literature (Li et al., 2010; Alonso et al., 2016; Bujkiewicz et al.; Buyse et al., 2000; Takagi & Kano, 2019). Specifically, (Takagi & Kano, 2019) showed the bias reduction when using surrogates outcomes. Therefore, surrogate outcomes can provide a promising way to resolve the missingness of primary outcome.

Surrogate outcomes should be handled with caution since they are post-treatment variables. Misusing them, for example, by including them as covariates, can lead to biased estimates of the treatment effects (Prentice, 1989; Athey et al., 2019; Cheng et al., 2020). We provide a motivating example to illustrate this in Section 3. In datasets with limited primary outcomes Kallus & Mao (2024) examined the role of surrogates and showed efficiency gains after including surrogate outcomes and unlabeled data in the analysis. Zeng et al. (2024) introduced a doubly robust method for estimating the average dose-response function using surrogate variables in the context of continuous treatments. In Liu et al. (2024), the information of the surrogate outcomes is adapted to the framework of proximal causal inference. Recently, Gao et al. (2025) exploited surrogate outcomes to conformal inference for the individual treatment effect. However, these methods either focus on the average treatment effect (ATE) estimation or did not provide a theoretical support for the CATE estimation.

We summerize the main contributions of this paper as follows:

- We introduce a Neyman-orthogonal framework for the CATE estimation in the presence of missing outcome and surrogate outcomes. We show that the loss function for CATE satisfies Neyman-orthogonal conditions Chernozhukov et al. (2018); Foster & Syrgkanis (2023), which shows that the CATE estimator based on this loss function is less sensitive to the nuisance parameters as the estimation errors of nuisance parameters is only of second order to the target parameter. This can produce more accurate and reliable results.

- We provide a theoretical foundation for the CATE estimator with surrogate outcomes. While existing theory only applies to compete data, we establish formal convergence guarantees under a MAR condition. Specifically, we prove that our CATE estimator converges to the true treatment effect function at oracle rate under a mild condition. The condition is sufficiently broad to accommodate flexible machine learning methods, including deep neural networks and random forests, for CATE estimation.

- The proposed estimation procedure can accommodate flexible methods to learn nuisance functions. We establish the convergence rate of the CATE estimator without additional structural restrictions on the nuisance functions beyond a consistency assumption with slow convergence rates. This model-agnostic feature enables the use of modern, off-the-shelf machine learning methods, which can handle complex prediction tasks while maintaining high practical accuracy.

## 2 RELATED WORKS

### 2.1 SEMI-SUPERVISED LEARNING

Our work contributes to the growing literature in semi-supervised learning, which contains both labeled and unlabeled outcomes. A substantial body of research has explored how unlabeled data can enhance the estimation of various parameters, including regression coefficients (Azriel et al., 2022; Hou et al., 2023), population means and ATEs (Chakrabortty & Dai, 2024; Zhang et al., 2023; 2019; Zhang & Bradic, 2021), ITEs (Harada & Kashima, 2020), as well as quantiles and quantile treatment effects (Chakrabortty et al., 2024). Most of these works assume, either implicitly or explicitly, that labels are MCAR. In contrast, we relax this assumption by allowing the labeling mechanism to depend on pre-treatment covariates, the treatment assignment, and even post-treatment variables: the surrogate outcomes. We emphasize the role of surrogates as an auxiliary source of information. Notably, the same framework can also be applied to cases when no surrogate outcomes are availalble.

## 2.2 CAUSAL INFERENCE WITH SURROGATE OUTCOMES

Numerous surrogate criteria have been proposed to ensure that treatment effects on surrogate outcomes can reliably predict the treatment effects on the primary outcome. The first criterion, introduced in Prentice (1989), requires the primary outcome to be conditionally independent of the treatment given the surrogate outcomes. Since then, many alternative criteria have been proposed, including the principal surrogate criterion (Frangakis & Rubin, 2004), strong surrogate criterion (Lauritzen et al., 2004), and consistent surrogate criterion (Chen et al., 2007; VanderWeele, 2013). While much of this literature focuses on a single surrogate, recent works by Price et al. (2018); Wang et al. (2019) estimated transformations of multiple surrogates to optimally approximate the primary outcome using labeled experimental data. Athey et al. (2019) explored identifying and estimating the ATE in a more complex setting, where the primary outcome and treatment are not observed in the same dataset. Subsequent works, such as Athey et al. (2020); Imbens et al. (2024), aimed to combine experimental short-term data with confounded observational long-term data. The former relies on a latent unconfoundedness assumption, while the latter uses multiple sequential surrogates as proxies. Similarly, Cai et al. (2024) designed a neural network architecture to combine experimental and observational data. Semiparametric inference for ATEs under the frameworks of Athey et al. (2019; 2020) were developed in Chen & Ritzwoller (2023). These works differ from ours as they use surrogates for identification. In contrast, our approach assumes that the primary outcome is MAR and uses surrogates to improve the CATE estimation in already-identified settings, which is close to the frameworks of Cheng et al. (2020); Kallus & Mao (2024).

## 2.3 CONDITIONAL AVERAGE TREATMENT EFFECT ESTIMATION

Our approach for CATE draws inspiration from Nie & Wager (2020), who cast the problem as a generic two-step loss minimization that can be implemented by off-the-shelf machine learning methods. The benefit of this decoupling is that it clearly separates the statistical tasks of estimating nuisance components from estimating treatment effects, which can be implemented and optimized (by standard cross-validation) through different machine learning algorithms. The final step of our approach takes the form of a pseudo-outcome regression, where transformed outcomes are regressed on covariates, and this approach dates back to van der Laan (2006); Luedtke & van der Laan (2016), who suggest it for estimating CATEs for complete data, but without explicit error guarantees. The error guarantee is provided in Kennedy (2023); Foster & Syrgkanis (2023); Curth & van der Schaar (2021) under general assumptions on the nuisance components (when estimated using sample splitting). They also derived theoretical properties for this approach to CATE estimation. This approach is extended in Sverdrup & Cui (2023) in the presence of unmeasured confounding.

**Notation.** We let $\xi_i$ represent Rademacher random variables. The Rademacher complexity of a function class $\mathcal{F} = \{f \mid f : \mathcal{X} \rightarrow \mathbb{R}\}$ is defined as $\mathrm{Rad}_n(\mathcal{F}) = \sup_{f \in \mathcal{F}} \left| \frac{1}{n} \sum_{i=1}^{n} \xi_i f(x_i) \right|$. For any two functions $f_1, f_2 \in \mathcal{F}$, we define the $L_\infty$-norm as $\|f_1 - f_2\|_\infty = \sup_{x \in \mathcal{X}} |f_1(x) - f_2(x)|$ and the $L^2$ norm as $\|f_1 - f_2\|_2 = \sqrt{\int_{x \in \mathcal{X}} |f_1(x) - f_2(x)|^2 \, dx}$. For a function class $\mathcal{F}$, we define $\|\mathcal{F}\|_\infty = \sup_{f \in \mathcal{F}} \|f\|_\infty$.

## 3 PROBLEM FORMULATION

Let $A \in \{0, 1\}$ be a binary treatment variable, $Y \in \mathbb{R}$ be an outcome of interest, and $X \in \mathcal{X} \subset \mathbb{R}^p$ be baseline covariates. Under the Neyman-Rubin potential outcome framework (Splawa-Neyman et al., 1990; Rubin, 1974), we assume that $Y(1)$ and $Y(0)$ are the potential outcomes of the treatment and control arm, respectively. The potential outcome $Y(a)$ is the outcome that would have been realized under each treatment option $A = a$. We also assume that the actual observed outcome is the potential outcome corresponding to the actual treatment, i.e., $Y = Y(A)$, which is the conventional consistency assumption in causal inference. Our goal is to estimate CATE, defined as

$$\tau(x) = \mathbb{E}[Y(1) - Y(0)|X = x].$$

CATE evaluates the heterogeneous treatment effects of treatment $A$ on the outcome $Y$ given the subject feature $X = x$. If $(X, A, Y)$ is fully observed, one could estimate the CATE from existing methods, such as (Hill, 2011; Alaa & van der Schaar, 2017; Hahn et al., 2020; Athey & Imbens,

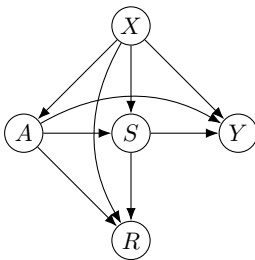

Figure 1: The causal DAG for the variables $(X, A, S, Y, R)$ illustrates the directional causal relationship among them, where each arrow represents the direction of causality.

2016; Wager & Athey, 2018; Johansson et al., 2016; Shalit et al., 2017; Yoon et al., 2018; Shi et al., 2019; Künzel et al., 2019; Nie & Wager, 2020; Kennedy, 2023) and the references therein.

For some individual $X$, the outcome $Y$ is missing. We denote $R$ the missing indicator, where $R = 1$ when $Y$ is observed, otherwise $R = 0$. In addition to $(X, A, Y, R)$, we also observe surrogate outcomes $S \in \mathcal{S} \subset \mathbb{R}^q$. We will present the results for $\mathcal{S} \neq \emptyset$ but note that $\mathcal{S} = \emptyset$ can be viewed as a special setting where our methodology still applies. Thus, the observations are: $(X_i, A_i, S_i, Y_i, R_i), i = 1, \cdots, n$, which are independent identically distributed copies of $(X, A, S, Y, R)$. We also denote $S(1)$ and $S(0)$ as the potential outcome of the surrogate outcome $S$. In this paper, we assume that $(X, A, S, Y, R)$ has the causal relationship in Fig 1, which is a causal DAG of (Pearl, 2009).

In summary, the sample $\mathbf{S}$ contains two subsets, a label subset $\mathbf{L} = \{Z_i = (X_i, A_i, S_i, Y_i, R_i = 1), i = 1, \ldots, n_l\}$ and an unlabel subset $\mathbf{U} = \{Z_i = (X_i, A_i, S_i, Y_i = \texttt{NA}, R_i = 0), i = n_l + 1, \ldots, n\}$, where NA stands for "Not Available", i.e., a missing value. Let the propensity score (PS) function be $\pi(x) = \mathbb{P}(A = 1 \mid X = x)$ and the observed probability be $\rho(x, a, s) = \mathbb{P}(R = 1 \mid X = x, A = a, S = s)$. To identify CATE, we need the following standard causal assumptions (Rosenbaum & Rubin, 1983) and missing at random assumptions Kallus & Mao (2024).

**Assumption 1.** *(a) Consistency:* $(S(a), Y(a)) = (S, Y)$ *almost surely when* $A = a$;
*(b) Ignoriability:* $Y(a) \perp A \mid X$ *for* $a = 0, 1$;
*(c) Positivity: there exist a constant* $c > 0$, *such that* $1 - c \geq \pi(x) \geq c$ *and* $\rho(x, a, s) \geq c$ *for all* $x \in \mathcal{X}$, $a \in \{0, 1\}$, *and* $s \in \mathcal{S}$;
*(d) Missing at Random:* $R \perp Y(a) \mid X, A, S(a)$, *for* $a = 0, 1$.

Assumption 1 requires the potential outcomes, for both the surrogate and primary outcome of an individual at the actual treatment $A$, be the same as the actual outcome of that individual. The ignorability assumption implies that there is no other confounders except for covariates $X$ that influence both the potential outcomes and the treatment assignment mechanism. The positivity assumption states that each individual has a positive chance of receiving treatment and has the primary outcome observed. The MAR assumption implies that the surrogate outcomes $S$ is informative to the primary outcome such that the distributions of labeled and unlabeled data are comparable after conditioning on $(X, A, S)$. These assumptions commonly hold in randomized experiments and well-designed observational studies.

It is worth noting that the MAR assumption above is considerably weaker than MCAR assumption and the MAR assumption in the previous literature (Chakrabortty & Dai, 2024; Zhang et al., 2023; Chernozhukov et al., 2018; Azriel et al., 2022; Hou et al., 2023; Zhang et al., 2019; Zhang & Bradic, 2021; Chakrabortty et al., 2024). Those assumptions restrict the missing mechanism to not depend on the surrogate outcomes $S(a)$. However, such assumptions may fail if the missing mechanism is predictable by the surrogate outcomes. For instance, it is possible that subjects with positive surrogate outcomes are more likely to drop out of the study as they expect themselves to be healthier and more likely to have positive primary outcomes. In this case, estimating CATE without considering the surrogate outcomes triggers a bias from an incorrect target population $\mathbb{E}[Y(1) - Y(0)|R = 1, X]$. Therefore, the surrogate outcomes are necessary for estimating CATE.

Incorporating surrogate outcomes in the estimation of CATE requires special handling, since they play a different role from the covariates and hence cannot be simply included in the model as a co-

variate. To demonstrate this, we consider a linear regression model with only one surrogate outcome and without missing:

$$Y = \alpha_0 + \alpha'_x X + \alpha_a A + \alpha_s S + \epsilon_y, \ \mathbb{E}[\epsilon_y \mid X, A, S] = 0$$
$$S = \beta_0 + \beta'_x X + \beta_a A + \epsilon_s, \ \mathbb{E}[\epsilon_s \mid X, A] = 0.$$

It is not difficult to verify that the CATE with respect to a covariate $X$ is $\tau(x) = \alpha_a + \alpha_s \beta_a$. However, if we regress $Y$ on $(X, A, S)$, we will get a biased estimate that targets $\alpha_a$ instead of $\alpha_a + \alpha_s \beta_a$. Such a phenomenon is analogous to the mediation analysis (Baron & Kenny, 1986; Robins & Greenland, 1992; Imai et al., 2010; VanderWeele, 2016). The effect of the treatment $A$ on the primary outcome $Y$ is mediated through the surrogate outcomes $S$. Regressing the primary outcome on both the treatment and the mediator leads to the biased estimator of treatment effects. After all, the surrogate outcomes are post-treatment variables and should not be treated as covariates when estimating the CATE.

## 4 METHODS

We first propose a novel method that can incorporate surrogate outcomes with any existing CATE estimators (Hill, 2011; Alaa & van der Schaar, 2017; Hahn et al., 2020; Athey & Imbens, 2016; Wager & Athey, 2018; Johansson et al., 2016; Shalit et al., 2017; Yoon et al., 2018; Shi et al., 2019; Künzel et al., 2019; Nie & Wager, 2020; Kennedy, 2023). We then use this idea to develop the Surrogate-Assisted Learner (SA-learner).

### 4.1 IMPROVEMENT USING SURROGATE OUTCOMES

Let $\mu(x, a, s) = \mathbb{E}[Y|X = x, A = a, S = s, R = 1]$ be the regression outcome of the observed data. We show the identification result utilizing the surrogate outcomes in Proposition 1.

**Proposition 1.** *Under Assumption 1, CATE is identifiable as:*

$$\tau(x) = \mathbb{E}_S[\mu(X, 1, S) \mid X = x, A = 1] - \mathbb{E}_S[\mu(X, 0, S) \mid X = x, A = 0],$$

*where $\mathbb{E}_S$ represents the conditional expectation taking over the surrogate outcome $S$ given $(X, A)$. For convenience, we denote $\mathbb{E}_S[\mu(X, A, S) \mid X = x, A = a]$ by $\nu(x, a)$ and its estimator by $\hat{\nu}(x, a)$.*

Proposition 1 suggests that we can use a two-step procedure to identify CATE when the primary outcome is missing. This motivates our approach to assist the CATE estimate by the surrogate outcomes. In the first step, we regress the primary outcome $Y$ on $(X, A, S)$ from the label data $\mathbf{L}$ and obtain the estimator $\hat{\mu}(x, a, s)$ for $\mu(x, a, s)$. We then evaluate it as a proxy of the primary outcome on the entire sample $\mathbf{S}$. In the second step, we regress the proxy $\hat{\mu}(X, A, S)$ on $X$ from the entire sample $\mathbf{S}$ for both the treated ($A = 1$) and the control ($A = 0$) groups. The CATE estimator is then obtained by taking the difference $\hat{\tau}(x) = \hat{\nu}(x, 1) - \hat{\nu}(x, 0)$. In fact, we can replace the second step by many CATE estimators for the complete dataset from the literature as the data is now completely imputed by the proxy $\hat{\mu}(x, a, s)$. We summarize the procedure in Algorithm 1 and illustrate it through the meta-learners (Künzel et al., 2019; Kennedy, 2023) in the Supplement and compare their numerical performance in Section 6.

**Algorithm 1.** *(CATE estimators with Surrogate outcomes)*

**Step 1.** *Train an appropriate machine learning algorithm of $\mu(x, a, s)$ on the label data $\mathbf{L}$ and get the evaluation on the entire data $\mathbf{S}$.*

**Step 2.** *Replace the primary outcome $Y_i$ by the proxy $\hat{\mu}(X_i, A_i, S_i)$, regardless of whether the primary outcome is observed or not, and apply the CATE estimation to the completed data $\{(X_i, A_i, \hat{\mu}(X_i, A_i, S_i)) : i = 1, \dots, n\}$ to obtain the CATE estimate $\hat{\tau}(x)$.*

Although Algorithm 1 offers an estimate for CATE when the primary outcome is not fully available, it is unsurprising that this CATE estimate is sensitive to the error in Step 1. To address such a concern, we propose the SA-learner, a doubly robust estimator, as a solution.

## 4.2  SA-LEARNER

We utilize the semiparametric theory to improve the CATE estimation. The idea is to find a pseudo-outcome $\zeta(z) := \zeta(z; \mu, \rho, \nu, \pi)$ depending on nuisance functions ideally with second-order dependence on nuisance estimation error such that $\mathbb{E}[\zeta(z; \mu, \rho, \nu, \pi)]$ is equal to the ATE. Following Robins et al. (1994); Robins & Rotnitzky (1995); van der Laan & Robins (2003); Tsiatis (2006), we consider the functional $\psi = \mathbb{E}[\tau(X)]$ under the MAR setting, which is pathwise differentiable and admits an efficient influence function. Then the pseudo-outcome $\zeta(z; \mu, \rho, \nu, \pi)$ is a component in the influence function of $\psi$. We omit the derivation of the influence function and only present the form of pseudo-outcome. Let $(\overline{\mu}, \overline{\rho}, \overline{\nu}, \overline{\pi})$ be some functions that may not necessarily be equal to the true $(\mu, \rho, \nu, \pi)$, and define a score functions

$$\zeta(z; \overline{\mu}, \overline{\rho}, \overline{\nu}, \overline{\pi}) = \overline{\nu}(x, 1) - \overline{\nu}(x, 0) + \varphi(z; \overline{\mu}, \overline{\rho}, \overline{\nu}, \overline{\pi}), \tag{1}$$

where

$$\varphi(z; \overline{\mu}, \overline{\rho}, \overline{\nu}, \overline{\pi}) = \frac{a - \overline{\pi}(x)}{\overline{\pi}(x)(1 - \overline{\pi}(x))} \left( \frac{r(y - \overline{\mu}(x, a, s))}{\overline{\rho}(x, a, s)} + \overline{\mu}(x, a, s) - \overline{\nu}(x, a) \right).$$

The corresponding semiparametric efficient ATE estimate is the sample average of $\zeta(Z_i; \overline{\mu}, \overline{\rho}, \overline{\nu}, \overline{\pi})$. The following proposition shows such a characterization of ATE through $\zeta(Z; \overline{\mu}, \overline{\rho}, \overline{\nu}, \overline{\pi})$.

**Proposition 2.** *Let $(\overline{\mu}, \overline{\rho}, \overline{\nu}, \overline{\pi})$ be nuisance functions that may not necessarily equal the true $(\mu, \rho, \nu, \pi)$. Assume that $(\overline{\rho}, \overline{\pi})$ satisfies the requirement of $(\rho, \pi)$ in Assumption 1(c). Then*

$$\mathbb{E}[\zeta(Z; \overline{\mu}, \overline{\rho}, \overline{\nu}, \overline{\pi})] = \psi$$

*if either $(\overline{\mu}, \overline{\nu}) = (\mu, \nu)$ or $(\overline{\rho}, \overline{\pi}) = (\rho, \pi)$.*

Proposition 2 implies the doubly-robustness of our method. It extends previous work for a complete dataset to the missing data setting. If the proxy perfectly represents the primary outcome, i.e., $\overline{\mu}(a, s, x) = y$ or the data is complete, i.e., $r = 1$ and $\overline{\rho}(x, a, s) = 1$, then the doubly robust score $\zeta(z; \overline{\mu}, \overline{\rho}, \overline{\nu}, \overline{\pi})$ reduces to the efficient influence function for complete data (van der Laan & Rose, 2011). The intuition is that, to efficiently estimate the ATE, the doubly robust estimator averages the pseudo-outcome $\zeta(Z; \overline{\mu}, \overline{\rho}, \overline{\nu}, \overline{\pi})$, so to estimate the CATE, it suffices to learn the mapping from covariates $X$ to the pseudo-outcome $\zeta(Z; \overline{\mu}, \overline{\rho}, \overline{\nu}, \overline{\pi})$. This motivates the following procedure:

**Algorithm 2.** *(SA-learner)*

**Step 1.** *We first split the data into $C$ equal-size folds, then estimate $\mu(x, a, s)$, $\rho(x, a, s)$, $\nu(x, a)$, $\pi(x)$ with cross-fitting over the $C$ folds, where $\hat{\mu}(x, a, s)$ is obtained from in Algorithm 1, and $\hat{\nu}(x, a)$ is form regressing $\hat{\mu}(x, a, s)$ on covariates $X$.*

**Step 2.** *Form Equation equation 1 using cross-fit plug-in estimates of nuisance components $\hat{\mu}^{(-c(i))}(x, a, s)$, $\hat{\rho}^{(-c(i))}(x, a, s)$, $\hat{\nu}^{(-c(i))}(x, a)$, $\hat{\pi}^{(-c(i))}(x)$, where the notation $c(\cdot)$ maps from sample to fold, and $(-c(i))$ indicates predictions without using the $i$-th sample for training. Let $\hat{\zeta}^{(-c(i))}(z) = \zeta(z; \hat{\mu}^{(-c(i))}, \hat{\rho}^{(-c(i))}, \hat{\nu}^{(-c(i))}, \hat{\pi}^{(-c(i))})$. We estimate the CATE by minimizing the following empirical loss*

$$\hat{\tau}(\cdot) = \arg \min_{\tau} \hat{L}(\tau), \tag{2}$$

*where*

$$\hat{L}(\tau) = \frac{1}{n} \sum_{i=1}^{n} (\hat{\zeta}^{(-c(i))}(Z_i) - \tau(X_i))^2. \tag{3}$$

We can leverage flexible non-parametric learners such as random forests and neural networks to get $\hat{\tau}(\cdot)$ in equation 2. An interesting conceptual connection is that under the completely observed outcome $Y$, Equation equation 1 reduces to the celebrated Augmented Inverse-Probability Weighted (AIPW) score (Robins et al., 1994; Robins & Rotnitzky, 1995), then $\hat{\tau}(\cdot)$ in equation 2 becomes a Doubly Robust Learner (DR-learner) (Kennedy, 2023).

We highlight that the empirical loss equation 3 in Algorithm 2 can be used for learning other estimates of interest with a minimal adjustment. For instance, we can also investigate the CATE on

the unlabeled (CATU), $\tau_{CATU}(x) = \mathbb{E}[Y(1) - Y(0)|R = 0, X = x]$. When the target is CATU, we are interesting in measuring the heterogenetic treatment effect among the missing subjects. Let $e(x) = \mathbb{P}(A = 1 \mid X, R = 0)$ be the PS function among the unlabeled. In this case, Assumption 1 (b) can be weakened by Ignoriability of the unlabeled: $Y(a) \perp A|X, R = 0$ for $a = 0, 1$. Assumption 1 (c) needs to be replaced by Positivity for both missing and observed: there exist a constant $c > 0$, such that $c \le e(x) \le 1 - c$ and $c \le \rho(x, a, s) \le 1 - c$ for all $x \in \mathcal{X}$, $a \in \{0, 1\}$, and $s \in \mathcal{S}$. Under the refined assumptions, we define the doubly robust score for the ATE on the unlabeled population as

$$\zeta_{CATU}(z; \overline{\mu}, \overline{\rho}, \overline{\nu}, \overline{e}) = \frac{1 - r}{\mathbb{P}(R = 0)}(\overline{\nu}(x, 1) - \overline{\nu}(x, 0)) + \varphi_{CATU}(z; \overline{\mu}, \overline{\rho}, \overline{\nu}, \overline{e}),$$

where

$$\varphi_{CATU}(z; \overline{\mu}, \overline{\rho}, \overline{\nu}, \overline{e}) = \frac{A - \overline{e}(x)}{\overline{e}(x)(1 - \overline{e}(x))}\left(\frac{1 - \overline{\rho}(x, a, s)}{\mathbb{P}(R = 0)}\frac{r(y - \overline{\mu}(x, a, s))}{\overline{\rho}(x, a, s)} + \frac{(1 - r)(\overline{\mu}(x, a, s) - \overline{\nu}(x, a))}{\mathbb{P}(R = 0)}\right).$$

To define the empirical loss, we replace the probability measure $\mathbb{P}$ by the empirical measure $\mathbb{P}_n$, and the nuisance functions $(\overline{\mu}, \overline{\rho}, \overline{\nu}, \overline{e})$ by their cross-fitted estimators $(\hat{\mu}^{(-c(i))}, \hat{\rho}^{(-c(i))}, \hat{\nu}^{(-c(i))}, \hat{e}^{(-c(i))})$. Let $\hat{\zeta}_{CATU}^{(-c(i))}(z) = \zeta_{CATU}(z; \hat{\mu}^{-c(i)}, \hat{\rho}^{-c(i)}, \hat{\nu}^{-c(i)}, \hat{e}^{-c(i)})$. The loss function is as follows

$$\hat{L}_{CATU}(\tau) = \frac{1}{n}\sum_{i=1}^{n}(\hat{\zeta}_{CATU}^{(-c(i))}(Z_i) - \tau(X_i))^2.$$

The rest of the learning procedure follows Algorithm 2. The theory below for CATE can be derived analogously.

## 5 THEORY

We present the rate of converge of the SA-learner using empirical processes theory (van der Vaart & Wellner, 1996). Similar to the empirical loss in Equation equation 3, we define the oracle loss function: $\tilde{L}(\tau) = \frac{1}{n}\sum_{i=1}^{n}(\zeta(Z_i) - \tau(X_i))^2$, and the oracle estimator: $\tilde{\tau} = \arg\min_{\tau \in \Gamma} \tilde{L}(\tau)$, where $\Gamma$ is a function space of the CATE.

We use $(\overline{\mu}, \overline{\rho}, \overline{\nu}, \overline{\pi})$ to denote fixed functions to which $(\hat{\mu}^{(-c(i))}, \hat{\rho}^{(-c(i))}, \hat{\nu}^{(-c(i))}, \hat{\pi}^{(-c(i))})$ converges to in the $L_\infty$-norm, i.e., $\|\hat{f} - \overline{f}\|_\infty = o_p(1)$, where $f$ represents the nuisance functions. We denote $\mathcal{U}$, $\mathcal{V}$, $\mathcal{P}$ and $\mathcal{Q}$ as the function space in which $\hat{\mu}^{(-c(i))}, \hat{\nu}^{(-c(i))}, \hat{\rho}^{(-c(i))}, \hat{\pi}^{(-c(i))}$ lies.

**Assumption 2.** *(a) There exists a constant $c$, such that $1 - c \ge \hat{\pi}^{(-c(i))}(x) \ge c$, $\hat{\rho}^{(-c(i))}(x, a, s) \ge c$, $\|\mathcal{U}\|_\infty < \infty$, and $\|\mu\|_\infty < \infty$; for all $(x, a, s) \in \mathcal{X} \times \{0, 1\} \times \mathcal{S}$, $\hat{\rho}^{(-c(i))} \in \mathcal{P}$ and $\hat{\pi}^{(-c(i))} \in \mathcal{Q}$,*
*(b) Either $(\overline{\mu}, \overline{\nu}) = (\mu, \nu)$ or $(\overline{\rho}, \overline{\pi}) = (\rho, \pi)$;*
*(c) For some constant $\gamma > 0$, the oracle estimator $\tilde{\tau}$ satisfies $\|\tilde{\tau} - \tau\|_2 = O_p(n^{-\gamma})$ with the corresponding function space $\Gamma$ satisfying $\mathrm{Rad}_n(\Gamma) = O(n^{-\eta})$ for some $0 < \eta \le 1/2$.*

Assumption 2 (a) requires the boundedness of the function spaces, which is standard for nonparametric regression. Assumption 2 (b) requires at least one of the pair, regression outcome estimation or the conditional probability estimation, be consistent. Such an assumption allows for model misspecification in the nuisance function estimation. Assumption 2 (c) concerns the rate of convergence of the oracle estimator. In the literature, the convergence rates have been extensively investigated. For example, the rate is of order $n^{-\alpha/(2\alpha+d)}$ for nonparametric regression (Wasserman, 2006) and of order $n^{-\alpha/(2\alpha+t)}\log^{3/2} n$ for a regularized ReLU neural network (Schmidt-Hieber, 2020), where $\alpha$ is the degree of smoothness of a $d$-dimensional true regression function in the CATE function space $\Gamma$, and $t \le d$ is the intrinsic dimension of the space $\Gamma$.

Before presenting the convergence rate of the SA-learner, we refine Proposition 2 in terms of the nuisance estimators $(\hat{\mu}^{(-c(i))}, \hat{\nu}^{(-c(i))}, \hat{\rho}^{(-c(i))}, \hat{\pi}^{(-c(i))})$.

**Proposition 3.** *Under Assumption 1 and 2, we can derive that*

$$|\mathbb{E}[\hat{\zeta}^{(-c(i))}(Z)] - \psi| = O_p(\max(r_\mu(n)r_\rho(n), r_\nu(n)r_\pi(n))),$$

*where $\psi$ is the ATE and $\|\hat{f} - f\|_\infty = O_p(r_f(n))$ with $f$ representing the nuisance functions $(\mu, \rho, \nu, \pi)$.*

Proposition 3 further characterizes the error from nuisance function estimation. The product terms $r_\mu(n)r_\rho(n)$ and $r_\nu(n)r_\pi(n)$ resemble the error terms associated with doubly robust scores in a complete dataset. Let $r(n) = \max(r_\mu(n)r_\rho(n), r_\nu(n)r_\pi(n))$. In the complete dataset, $\mu(x, a, s) = y$ and $\rho(x, a, s) = 1$; thus $r_\mu(n) = r_\rho(n) = 0$. The error term $r(n)$ reduces to $r_\nu(n)r_\pi(n)$, which is identical to the known error bound for the doubly-robustness in CATE estimation when there is no missing data (Kennedy, 2023). We are now ready to present the convergence rate of the SA-learner.

**Theorem 1.** *Under Assumption 1 and 2, we have*

$$\|\hat{\tau} - \tau\|_2 = O_p(n^{-\gamma} + r(n)).$$

*Furthermore, if $r(n) > n^{-\gamma}$, then $\|\hat{\tau} - \tau\|_2 \asymp \|\tilde{\tau} - \tau\|_2$.*

Theorem 1 ensures that, by suitably controlling model complexity and under some mild assumptions on the nuisance estimators, the SA-learner is doubly robust in the sense that as long as one of the pair of the nuisance function estimations is consistent, then the SA-learner is also consistent. It also implies that the cross-fitted SA-learner can attain performance comparable to that of the oracle learner, which has prior knowledge of all nuisance functions $(\mu, \rho, \nu, \pi)$. Moreover, when all nuisance function estimators are consistent, the SA-learner converges to the truth at a rate faster than the rates of the nuisance estimators. Therefore, employing the SA-learner theoretically leads to a better estimator of the CATE. The proof of Theorem 1 is given in the Supplement.

# 6 EXPERIMENTS

For empirical evaluation, we conduct two experiments and follow prior work on treatment effect estimation to examine the performance on synthetic dataset.

**Datasets.** In the first dataset, we consider a simple MCAR setting such that the missing rate is approximately 50%. We adapt the mean function from Györfi et al. (2002) and the treatment mechanism from Kennedy (2023). The synthetic data contains 1000 observations with one covariates and two surrogate outcomes. The details of the simulation are provided in the Appendix. In the second dataset, we construct a MAR setting such that the missingness mechanism is conditionally independent of the primary outcome $Y$, given the surrogate outcomes $S$. The marginal missing rate is about 30%. For the rest of settings, We follow the simulation of "Setup A" in Wager & Athey (2018). The synthetic dataset are generated across three different sample sizes: $n = 1000, 2000, 3000$ with 5 covariates and 2 surrogate outcomes. Again, the details of the simulation are provided in the Appendix.

**Baseline Methods.** We compare the performance of the SA-learner to four well-established meta learner algorithms: S-learner (Künzel et al., 2019), T-learner (Künzel et al., 2019), X-learner (Künzel et al., 2019) and DR-learner (Kennedy, 2023). Since these four baseline meta-learners, are designed for complete data and cannot handle missing values, we exclude the observations with missing outcomes and train the baseline methods solely on the labeled sample **L**.

**Implementation Details.** To demonstrate the performance of two estimators under varying nuisance estimation errors, we will manually assign the estimation error in the first dataset, which is suitable for simulation purposes. For a fixed $\alpha > 0$, we set $\hat{\mu} = \mu + N(1, 1)$, $\hat{\nu} = \nu + N(1, 1)$, $\text{logit}(\hat{\rho}) = \text{logit}(\rho) + N(n^{-\alpha}, n^{-2\alpha})$, and $\text{logit}(\hat{\pi}) = \text{logit}(\pi) + N(n^{-\alpha}, n^{-2\alpha})$ so that $RMSE(\hat{\rho}) \approx RMSE(\hat{\pi}) \approx n^{-\alpha}$, and the error rate of $(\hat{\rho}, \hat{\pi})$ is dominated than that of $(\hat{\mu}, \hat{\nu})$. In this case, S-learner and T-learner are analogous as a plug-in estimator so we only present the T-learner in the first dataset. In the second dataset, we also implement Algorithm 1. Algorithm 1 takes 4 different Baseline Methods as its default learners. We employ a flexible machine learning model to estimate the nuisance functions, but use a simple linear regression to estimate the CATE. For estimation of the nuisance function, the outcome models, such as $\mu(x, a, s)$ and $\nu(x, a)$, are implemented using XGBoost; while the probability models, such as $\rho(x, a, s)$ and $\pi(x)$, are implemented using logistic regression. All methods are trained and evaluated using cross-validation in each dataset.

**Metrics.** We measure the precision in the estimation of heterogeneous effect (PEHE) by $\epsilon_{PEHE} = \sqrt{\frac{1}{n}\sum_{i=1}^n (\hat{\tau}(X_i) - \tau(X_i))^2}$, and visualize the averaged PEHE across 200 replicates in Figure 2a and Figure 2b.

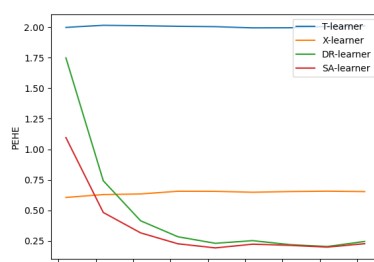 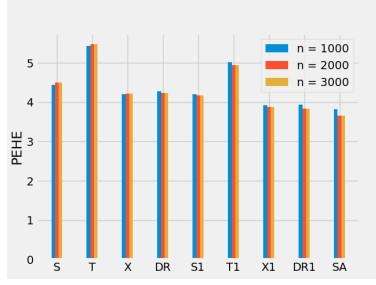

(a) PEHE with respect to the rate of conver- (b) PEHE for selected meta-learners
gence from the nuisance function estimation across different sample sizes

Figure 2: Simulation results

**Result.** From Figure 2a, the results indicate that the plug-in estimator, T-learner, inherits the large errors from estimating the individual regression functions, whereas the SA-Learner achieves substantially smaller errors and adapts to the smoothness of the CATE. The X-Learner attains an MSE that lies between the two. The DR-Learner exhibits a trend similar to that of the SA-Learner, but with a higher PEHE and a slower convergence rate, possibly due to sample inefficiency. Consistent with Theorem 1, the MSE of the SA-Learner approaches that of the oracle as the propensity score estimation error diminishes (i.e., as the convergence rate increases).

From Figure 2b, we observe that all meta learners utilizing surrogate outcomes outperform those that do not, for instance S versus S1. This confirms the effectiveness of Algorithm 1 for the benefits of surrogate outcomes. Moreover, the SA-learner performs best among all methods. The relative performance of the SA-Learner appears to improve as the sample size increases.

# 7 CONCLUSION

This paper introduces the SA-Learner, a novel method for estimating heterogeneous treatment effects in the presence of missing outcomes. By leveraging surrogate outcomes, the SA-Learner effectively addresses the challenges of bias and efficiency loss commonly encountered in real-world data with missing responses. The SA-Learner enjoys double robustness, ensuring consistent CATE estimates even under misspecification of certain nuisance functions. Additionally, we also establish its convergence rate, requiring only slower convergence rate for the nuisance function estimators without restrictive model assumptions. This property enables flexible implementation using off-the-shelf machine learning algorithms. Through extensive experiments on synthetic data, we empirically validates the effectiveness of the proposed method and demonstrates its superiority over competing meta-learners. Our methods thus constitute valuable additions to the CATE estimation toolkit. Their broader impact will likely be to improve estimation accuracy in existing HTE applications.

In the future, practical adaptations of the SA-learner may be explored to accommodate multiple and continuous treatments. Multiple treatments arise in various applications; for example, waiting time before follow-up, percent of discount in marketing studies, and drug dosage in clinical trials (Imai & van Dyk and, 2004; Hirano & Imbens, 2004; Bretz et al., 2005; Cattaneo, 2010). Analyzing multiple treatments provides valuable insights into causal effects across different treatment levels but poses great challenges for CATE estimation, as additional assumptions are required for identification. Recently, Acharki et al. (2023) extended the meta-learner methods to the multiple-treatment setting. Therefore, a natural future direction is to extend our SA-learner to this context. Another direction is to extend the framework to the Missing Not At Random (MNAR) setting. Strictly speaking, our MAR setting corresponds to an MNAR scenario in the classical causal inference framework, as missingness may depend on external randomness through surrogate outcomes. Nonetheless, concerns about potential unmeasured confounding may still be raised. A potential solution is to leverage tools from proximal causal inference to address unmeasured confounders associated with the missingness (Liu et al., 2024; Sverdrup & Cui, 2023; Cui et al., 2024; Mastouri et al., 2023).

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

## A   APPENDIX

The first dataset is simulated as follows:

$$X_i \sim U(-1, 1), \ A_i \sim \text{Bernoulli}(0.5 + 0.4 \, \text{sign}(X_i)),$$
$$S_i \sim N_2(0, I_2), \ R_i \sim \text{Bernoulli}(0.5),$$
$$Y_i = b(X_i, A_i, S_i) + A_i \tau(X_i) + \epsilon_i(X_i), \epsilon_i(X_i) \sim N(0, (0.2 - 0.1 \cos(2\pi X_i))^2),$$

where the base line function is $b(X_i, A_i, S_i) = \mu(X_i) + A_i \tau(X_i) + 0.1 S_{i1} - 0.1 S_{i2}$ with

$$\mu(x) = \begin{cases} (x+2)^2/2 & \text{if } -1 \le x < -0.5; \\ x/2 + 0.875 & \text{if } -0.5 \le x < 0; \\ -5(x-0.2)^2 + 1.075 & \text{if } 0 \le x < 0.5; \\ x + 0.125 & \text{if } 0.5 \le x \le 1, \end{cases}$$

and the underlying CATE function is $\tau(X_i) = 1$. Note that the observed indicator $R_i$ is independent of $Y_i$.

Next, we generate the second dataset. Let $\text{trim}_\eta(x) = \max(\eta, \min(x, 1 - \eta))$ and $\text{sigmoid}(x) = 1/(1 + e^{-x})$. We have

$$X_i \sim U(0, 1)^5, \ A_i \sim \text{Bernoulli}(\text{trim}_{0.1}(\sin(\pi X_{i1} X_{i2}))),$$
$$S_i \sim N_2((1 - 2A_i)\mathbf{1}, I_2), \ R_i \sim \text{Bernoulli}(\text{sigmoid}(S_{i1}/2 + S_{i2}/2 + 1)),$$
$$Y_i = b(X_i, A_i, S_i) + (A_i - 0.5)\tau(X_i) + \epsilon_i, \epsilon_i \sim N(0, 1),$$

where the base line function is $b(X_i, A_i, S_i) = \sin(\pi X_{i1} X_{i2}) + 2(X_{i3} - 0.5)^2 + X_{i4} + 0.5 X_{i5} + (1 - 2A_i)(S_{i1} + S_{i2})$, and the underlying CATE function is $\tau(X_i) = (X_{i1} + X_{i2})/2$. Note that the observed indicator $R_i$ is conditional independent of the primary outcome $Y_i$ given the surrogate outcomes $S_i$.

