# OpenReview forum: "SA-Learner: Surrogate-Assisted Meta-Learner with Missing Outcomes"
_ICLR.cc/2026/Conference — ICLR 2026 Conference Withdrawn Submission_

### Official Review · Reviewer_wHbY · 2025-10-15

**Soundness:** 3
**Presentation:** 3
**Contribution:** 1
**Rating:** 2
**Confidence:** 4

**Summary:**

The authors propose a new orthogonal meta-learner for estimating conditional average treatment effects (CATEs) in a semi-supervised setting with outcomes missing at random and observed surrogates. The key idea is to leverage the efficient influence function to construct a doubly robust pseudo-outcome similar as done in existing works on estimating CATEs. The authors also propose to learn CATE conditioned on the unlabeled population (called CATU) using a similar idea. Finally, the resulting learners are validated using synthetic data.

**Strengths:**

- Neyman orthogonal learners are established in causal inference literature and are known for several favorable properties. The proposed learners fall within this framework and share these properties
- The existing experimental results seem promising
- Missing outcomes are common in applications, thus

**Weaknesses:**

- Novelty: It is well known in causal inference that Neyman orthogonal learners can be easily constructed once the Efficient influence function of the target quantity is known. The efficient influence functions used in this paper are, to the best of my knowledge, known in the literature. As such, the contribution of this paper reduces to the construction of orthogonal learners, which follows the established standard framework and limits novelty. Also, the theoretical results follow from this general framework and essentially leverage the properties of the known EIFs.
- Related work: The authors draw connections to surrogacy causal inference literature and semi-supervised learning, but I think that key works are overlooked. For example, prediction-powered inference (Angelopoulos et al.) and various follow-up works consider a very similar setting and also leverage semiparametric inference. In this literature, there are also existing approaches for learning CATE-like functions (i.e., considering estimation for conditional averages instead of the mean), e.g., "Doubly Robust Self-Training". I recommend a careful and more comprehensive literature search. Finally, the following paper considers a very similar setting to CATU: "A Meta-learner for Heterogeneous Effects in Difference-in-Differences" (Example 4.2).
- The experiments are relatively limited. It would be nice to see some more comprehensive experiments (e.g., a larger variety of surrogate strengths, or generally of the DGP). It would also be nice to add a case study using real-world data to demonstrate the applicability of the proposed learners.

**Questions:**

- Could the authors ellaborate on their contribution as compared to existing works?

---

### Official Review · Reviewer_aLnq · 2025-10-20

**Soundness:** 3
**Presentation:** 3
**Contribution:** 1
**Rating:** 2
**Confidence:** 3

**Summary:**

This work introduces the surrogate assisted (SA) learner, which is a doubly-robust learner for estimating CATE when outcomes are missing at random (SAR), but surrogate outcomes are available. Ignoring samples with missing outcomes can lead to bias, and hence leveraging surrogates is important to avoid this source of data bias. The SA learner improves upon this naive approach and provides learning guarantees with flexible machine learning methods.

**Strengths:**

- Complete piece of work: The paper is self-contained, as it provides all the necessary notation, assumptions, and proofs.

- Doubly robustness, convergence rates: The theoretical insights are precise and insightful. Doubly-robustness and quasi-oracle efficiency are a very desirable properties.

- Writing: The paper is very well written and easy to follow.

- Technical soundness: While I did not carefully check all the proofs line by line, the paper seems to be technically correct.

**Weaknesses:**

While there are only two weaknesses that are apparent to me, they are very important:

- **Limited contribution:** While there is some research gap that is filled with this work, the contribution is rather limited. Previous works have already established such learners for the ATE setting; the work presented here basically boils down to **deriving a rather trivial influence function and regressing it on covariates.**

- **Insufficient empirical validation:** The experiments section is insufficient. To benchmark a novel model-agnostic learner, the authors should provide several experimental setups. Instead, there is only a single dataset with a single metric on 3 different sample sizes. This is clearly **not enough**. The authors should explore different datasets, missingness rates, machine learning models, etc.

**Questions:**

Questions:
- What is the technical novelty of the SA-learner over surrogate-assisted ATE learners? For the latter, the EIF is (almost) the same.
- Regarding experiments, have the authors considered evaluating their SA learner with different machine learning models?
- Why is the SA-learner validated on only a single datasets with three very similar setups?
- Have the authors considered evaluating their method on different levels of missingness?

---

### Official Review · Reviewer_13Lx · 2025-10-25

**Soundness:** 1
**Presentation:** 1
**Contribution:** 1
**Rating:** 2
**Confidence:** 4

**Summary:**

The paper proposes a new CATE learner in the setting with data missingness. The proposed CATE learner in missing data setting is doubly robust.

**Strengths:**

The authors are the first to write down the form of the Doubly robust CATE learner for the setting of data missingness.

**Weaknesses:**

- The efficient influence function (EIF) on which the SA learner is based is not derived in the paper, only stated. It is the understanding of the reviewer that this EIF has been derived in previous work by Kallus & Mao (2024) ([https://arxiv.org/pdf/2003.12408](https://arxiv.org/pdf/2003.12408)).
- The method by which the SA-learner is constructed is standard in debiased ML/double ML/orthogonal statistical learning literature. While extending this approach to a new setting is a novel contribution, it is the view of the reviewer that the difficulty lies in the EIF derivation, which the review understands to already exist in the literature (Kallus & Mao 2024). Additionally, the mechanism of the existing framework is not presented clearly.
- Propositions 2 and 3 are statements about the ATE estimator implied by the CATE estimator. It is the understanding of the reviewer that existing work (Kallus & Mao 2024) has derived an efficient doubly robust estimator for the ATE in the data missingness setting.  If the claimed gap in literature concerns CATE estimators only, then these propositions are irrelevant.
- The first claimed contribution is demonstrating neyman-orthogonality of the CATE loss. This claim is not mentioned again anywhere in the paper. Keywords "Neyman-orthogonal" or just "orthogonal" are mentioned in the entire paper only once, in the claimed contribution. It is understanding of the reviewer that neyman-orthogonality falls out directly as a result of how the sa-learner is constructed, however, this is never mentioned by the authors (and the construction method is not novel to the paper). If such weight is placed on neyman orthogonality, we would expect the result to be emphasized more than as Lemma 2 in Proof of Theorem 1 in the Supplements.
- The paper is badly written with a lot of typos. For example,
	- 1. Line 38, the "and meta-learners (Ku ̈nzel et al., 2019; Nie & Wager, 2020; Kennedy, 2023), assumes" should be assume.
	- 3. Line 52, "We show in this paper that it is possible to provide further improvements Under the MAR settings" the under should be lower case.
	- 4. Line 74, "However, these methods either focus on the average treatment effect (ATE) estimation or did not provide a theoretical support for the CATE estimation." the "focus" and "did" should be in the same tenses.
	- 5. Line 306, "Form Equation equation 1" should be "from equation 1".
	- 6. Line 422, the "Baseline Methods" shoule be lower case.

Beyond all the typos present throughout the paper, the paper does not correctly use standard statistical terminology. For example, zeta (equation 1) is not a score function.

**Questions:**

- Did the authors have to derive a new efficient influence function? If yes, could they provide this derivation?
- Beyond the change of focus from ATE to CATE, is there a difference in the underlying setting from existing work of e.g. Kallus & Mao 2024 ([https://arxiv.org/pdf/2003.12408](https://arxiv.org/pdf/2003.12408)), who already derive the EIF?
- Are propositions 2 and 3 related to the proposed CATE estimator, or only about the implied ATE estimator? If they are statements relating only to implied ATE estimator, how are they different from existing work on ATE estimation in the data missingness setting?
- Could the author provide more experiment detail? For example, the data-generating process of covariates and treatment? And the experiments considered X with only one covariate and implemented with XGBoost and logistic regression, which does not very related with real-world problem. I would expect more experiments on more complex settings.

---

### Official Review · Reviewer_NrS3 · 2025-10-26

**Soundness:** 3
**Presentation:** 2
**Contribution:** 2
**Rating:** 2
**Confidence:** 3

**Summary:**

This paper addresses the problem of estimating heterogeneous treatment effect when there are missing outcomes. Instead of leveraging surrogates for identification, this work considers the case where the outcome is missing at random and leverages the covariate information to improve the estimate. The authors proposes a doubly robust pseudo-outcome for the ATE and can estimate the CATE by regressing the covariate on the pseudo-outcome. In addition, the paper also discusses estimation of treatment effects on the population with missing data.

**Strengths:**

- The related works section provides a good overview of different research directions in causal effect estimation with missing data.
- The proposed framework is flexible and the nuisance functions can be estimated using any ML models.

**Weaknesses:**

- Theoretical claims in this paper seems to follow the methodologies from prior works, with limited novelty.
- Results for the proposed method seems comparable with the DR-learner, which does not show significant improvement.
- There are some typos/missing punctuation that can be fixed.

**Questions:**

- Since the model introduces additional nuisance models to account for data missingness and incorporating surrogates, it seems like there would be a tradeoff between the errors from the additional models and the efficiency gain from incorporating new data. In there a way to maybe quantify the order of the proportion of missing data needed for the proposed method to be advantageous?

---

### Note · Authors · 2025-11-28

I have read and agree with the venue's withdrawal policy on behalf of myself and my co-authors.